# WaSCR: A WebAssembly Instruction-Timing Side Channel Repairer

## Abstract

WebAssembly (Wasm) is a platform-independent, low-level binary language that enables near-native performance in web applications. Given its growing importance in the web ecosystem, securing WebAssembly programs becomes increasingly important. A key security concern with WebAssembly is the threat of instruction-timing side-channel attacks, which exploit timing variations in branch instructions dependent on sensitive data, allowing attackers to infer sensitive information through timing measurement.

In this paper, we introduce WaSCR, an automated **W**eb**A**ssembly instruction-timing **S**ide-**C**hannel **R**epairer. WaSCR uses control and data dependencies to trace the flow of sensitive data and prevent its leakage. It employs rule-based code transformations to linearize the program, eliminating branches dependent on sensitive data and substituting them with constant-time selectors. Our evaluation demonstrates that WaSCR effectively eliminates instruction-timing side channels while maintaining program correctness, with efficient repairs and moderate performance overhead.

## 1 Introduction

WebAssembly (Wasm) is a platform-independent, low-level binary language designed to enable near-native performance in web applications [58]. It is widely supported by major browsers [33] and increasingly popular in the web ecosystem [26]. Despite operating within a sandboxed environment, which is generally considered secure [58], WebAssembly remains vulnerable to side-channel attacks [3, 28, 55, 56]. These attacks exploit a program's non-functional properties, such as execution time, cache access, or power consumption, to infer sensitive information like passwords or encryption keys [13, 16, 29, 35]. Research has shown that side-channel vulnerabilities can be exploited in WebAssembly programs [55], and existing protections have proven inadequate [56].

One common form of side-channel attacks is the *instruction-timing* attack, where an attacker deduces sensitive information by measuring the execution time of instructions in conditional branches. However, most WebAssembly side-channel research has focused on other types of attacks, such as Spectre [32, 36, 53], port contention [43], and cache attacks [14, 23], with limited work on addressing instruction-timing side channels. Tools developed for other languages, such as C/C++ and Java [12, 18, 34, 47, 48, 52, 59] – including those based on LLVM – cannot be easily adaptable to WebAssembly due to its unique language features and the diversity of its runtimes, such as V8 [50] and Wasmtime [4], which employ various compiler infrastructures.

Current solutions for mitigating instruction-timing side channels in WebAssembly are limited to CT-Wasm [17, 54] and the work by Tsoupidi et al. [51]. CT-Wasm extends WebAssembly's semantics to enforce constant-time programming [10], ensuring that program execution time is independent of sensitive data by introducing a "secret" data type and prohibiting its use in conditional branches.

However, since CT-Wasm is not part of the standard WebAssembly specification, it is implemented as an extension to the WebAssembly reference interpreter and V8 JavaScript engine. Similarly, Tsoupidi et al. used Relational Symbolic Execution (RelSE) to detect constant-time violations in WebAssembly, but their approach also requires modifications to the WebAssembly reference interpreter.

While existing approaches [51, 54] provide some protection against instruction-timing attacks in WebAssembly, they have significant limitations. First, they require platform-specific extensions, which limit their portability across platforms. Second, although these tools can detect constant-time violations in WebAssembly programs, fixing these violations still requires substantial manual effort from developers, making the process labor-intensive and error-prone. Consequently, there is an urgent need for robust, automated defenses to effectively detect and repair instruction-timing side channels in WebAssembly.

However, repairing instruction-timing side channels in WebAssembly presents distinct challenges. First, WebAssembly's unique type system, stack-based architecture, and memory model make existing defenses from other languages ineffective. Second, the absence of high-level data types and semantic metadata complicates the tracking of dependencies within WebAssembly's linear memory. Third, WebAssembly's diverse control structures and branching mechanisms increase the complexity of accurately modeling and transforming programs. Additionally, WebAssembly blocks can return values, adding another layer of intricacy that requires sophisticated analysis to effectively monitor data flow and taint propagation across different blocks. Finally, indirect calls — dynamic function invocations based on a function table — further hinder analysis and program transformation efforts.

To address these challenges, we introduce WaSCR, a static analysis tool designed to automatically detect and repair instruction-timing side channels in WebAssembly programs. Figure 1 illustrates WaSCR's architecture. It takes a WebAssembly module, along with a list of functions and user-annotated sensitive data, as input. It constructs a Program Dependency Graph (PDG) [21] for the WebAssembly module, on which it then performs taint analysis. This analysis traces both data and control dependencies to identify branches and code blocks affected by sensitive data, marking them as vulnerable. Upon identifying these vulnerabilities, WaSCR applies predefined rule-based code transformations. This process linearizes the vulnerable branches using constant-time selectors, ensuring the WebAssembly module's execution remains independent of sensitive data. This approach effectively mitigates instruction-timing side channel vulnerabilities.

We evaluate WaSCR on 20 WebAssembly modules across three key dimensions. First, we demonstrate the *effectiveness of WaSCR* in repairing instruction-timing side-channel vulnerabilities using GEM5 [11, 31], a fine-grained CPU architecture simulator. Second, we illustrate WaSCR's *efficiency of the repair process* by measuring

Figure 1: Overview of WaSCR

```
1  (module
2    (memory (export "memory") 1)
3    (func (export "add_and_store")
4      (param $p i32) (param $q i32) (local $a i32)
5      local.get $p
6      local.get $q
7      i32.add
8      local.set $a
9      i32.const 0 ;; memory address to store
10     local.get $a ;; value to store
11     i32.store offset=0 ))
```

(a) WebAssembly code `module.wasm`

```
1  const { instance } = await WebAssembly.
       ↪ instantiateStreaming(fetch('module.wasm'));
2  instance.exports.add_and_store(20);
3  const memory = new Uint32Array(instance.exports.
       ↪ memory.buffer);
4  console.log(memory[0]);
```

(b) JavaScript glue code

Figure 2: Example of Wasm and its JS glue code

the time required for leakage detection and code transformation. Third, we evaluate the *quality of the repaired programs* by measuring runtime overhead and code size increase. Our results confirm that WaSCR effectively mitigates instruction-timing vulnerabilities, faithfully preserves semantic correctness, and introduces moderate overhead.

In summary, our work makes the following contributions:

- We introduce WaSCR, a static analysis tool that automatically detects and repairs instruction-timing side channels in WebAssembly without platform-specific extensions.
- WaSCR uses taint tracking and rule-based code transformation to identify and linearize sensitive conditional branches in WebAssembly, mitigating timing vulnerabilities.
- We evaluate WaSCR on 20 WebAssembly modules, demonstrating its effectiveness in mitigating instruction-timing side channels while ensuring efficiency and quality.

## 2 Background and Motivation

This section provides an introduction of WebAssembly, explains how instruction-timing side-channel attacks work, and outlines the use of constant-time selectors to mitigate such vulnerabilities.

### 2.1 WebAssembly

WebAssembly is a low-level, platform-independent binary language that executes in a sandboxed environment [58]. It can be compiled from high-level languages such as C/C++, Rust, and Go [24]. Its ability to deliver near-native performance has made it increasingly popular, particularly in web applications where it runs alongside JavaScript. Figure 2 illustrates a WebAssembly module along with its JavaScript glue code. Specifically, the WebAssembly function add_and_store (Figure 2a) adds the two parameters, stores the result in a local variable $a, and writes the value of $a to memory address 0. The JavaScript glue code (Figure 2b) initializes the WebAssembly module, invokes the add_and_store function, and prints the addition result.

WebAssembly distinguishes itself from other programming languages with several key features:

**Strict Type System.** Unlike high-level languages that support a wide range of data types and automatic type inference, WebAssembly enforces a strict type system with only four numeric data types (i32, i64, f32, and f64). These types must be explicitly defined, making it resemble low-level machine code.

**Stack-based Virtual Machine.** WebAssembly operates on a stack machine for instruction execution, instead of using registers or memory for computation, as seen in most high-level languages. As shown in Figure 2a, values are pushed to the stack using local.get, consumed by i32.add, and the result is pushed back onto the stack.

**Linear Memory.** It uses linear memory, a continuous memory region that both WebAssembly code and host environments (e.g., JavaScript) can access directly. This differs from high-level languages that abstract memory through automated mechanisms like garbage collection. Although native assembly also interacts directly with memory, WebAssembly's memory model is more constrained, adding an extra security layer through sandboxing.

These distinctive features make analyzing WebAssembly fundamentally different from high-level languages and native assembly code. While WebAssembly is designed with security in mind, side-channel vulnerabilities remain a significant concern and require specialized analysis and targeted mitigation techniques.

### 2.2 Instruction-Timing Side Channels

Figure 3 shows a WebAssembly function that compares a password with a guessed input sequence, potentially exposing sensitive password information through instruction-timing side channels. The function's parameters represent the starting memory addresses of the password and the guessed sequence. To highlight the core

```
1  (func $check_password
2  (param $passwd_ptr i32) (param $guessed_seq_ptr i32) (result i32)
3  (local $ret_val i32)
4      ...
5      loop  $loop
6          ... ;; load the characters from the 2 strings
7          local.get $pwd_i
8          local.get $guess_i
9          i32.eq
10         local.set $cond
11         block $cmp
12             local.get $cond
13             br_if $cmp ;; continue $loop if $pwd_i == $guess_i
14             i32.const -1
15             return ;; return -1 if $pwd_i != $guess_i
16         end
17         ...  ;; continue $loop
18     end
19     i32.const 0
20     return ;; return 0 if all characters match
21 )
```

**Figure 3: Example of instruction-timing side channels**

```
1  i32.const 11              1  movl rcx,0x7
2  i32.const 7               2  movl rdx,0xb
3  local.get $p              3  # test the condition:
4  ;; select 7 if $p is false, 4  testl rax,rax
       ↪ otherwise select 11  5  # conditional move:
5  select                    6  cmovzl rdx,rcx
```

**(a) select in WebAssembly**   **(b) select in x86-64 code**

**Figure 4: Example of the WebAssembly select instruction**

logic responsible for the side-channel leak, we present a simplified version of the code.

In this example, the function $check_password iteratively loads and compares each character of the password and the guessed sequence, storing the result of each comparison in the variable $cond. Depending on this comparison result, the function jumps to different branches within the $cmp block. If the two characters match, the function breaks out of the $cmp block and continues the loop iteration. If the loop completes without a mismatch, the function returns with value 0. If a mismatch occurs, the loop terminates early, returning the function with value -1. It is important to note that in WebAssembly, when br_if targets blocks, it breaks to the end of the block when the condition is true. In contrast, when targeting loops, br_if branches to the beginning of the loop under the true condition.

This branching behavior leads to varying execution times based on where the mismatch occurs, creating a timing discrepancy. Consequently, an attacker can infer the correct password by carefully measuring execution times across different input sequences.

## 2.3 Repair with Constant-Time Selectors

To mitigate instruction-timing side channels, one widely-use approach is linearizing program branches using constant-time selectors [12, 47, 48, 52, 59]. A constant-time selector is a technique used in programming to ensure that an action takes the same amount of time regardless of input conditions. Its purpose is to prevent attackers from inferring sensitive information (e.g., cryptographic keys or passwords) by measuring program execution time. In WebAssembly, the select instruction acts as a constant-time selector when executed on runtime engines such as V8 [50] and Wasmtime [4]. These engines implement select using the CMOVcc conditional move instructions in x86-64 architecture. In contrast, branch instructions such as JZ in x86-64 can be affected by CPU features like speculative execution and branch misprediction, making them

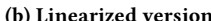

```
1  block $blk                    1  local.get $stack_ptr
2    local.get $sens_data        2  i32.load offset=12
3    br_if $blk                  3  local.set $prev
4    local.get $__stack_pointer  4  ;; select prev or new:
5    i32.const -1                5  local.get $__stack_pointer
6    i32.store offset=12         6  local.get $prev
7  end                           7  i32.const -1
                                 8  local.get $sens_data
                                 9  select ;; ct-selector
                                 10 i32.store offset=12
```

**(a) Sensitive-data-dependent flow**   **(b) Linearized version**

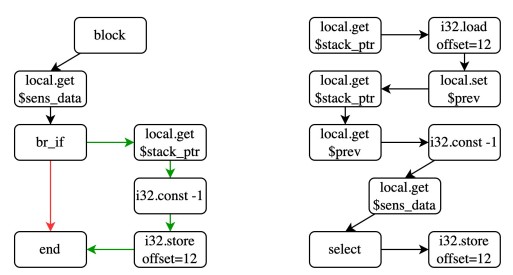

**(c) Contro flow of Figure 5a**   **(d) Control flow of Figure 5b**

**Figure 5: Example of branch linearization**

ineffective at eliminating timing variations. Similarly, directly using if/else WebAssembly instructions cannot guarantee constant-time behavior, even if both branches are equalized. Instead, CMOVcc instructions provide a reliable constant-time alternative for conditional selection, a characteristic that has been validated by existing research [12, 59]. Figure 4 shows how the select instruction is implemented in WebAssembly and its corresponding x86-64 machine code, compiled with Node.js version 20.13. The constant-time selector allows us to linearize sensitive-data-dependent branches, thereby mitigating potential instruction-timing side channels.

Figure 5 illustrates how branches can be linearized using constant-time selectors. Initially (Figure 5a), the execution flow is sensitive to the value of $sens_data, as seen in the conditional branching (line 3). If $sens_data is true, the program breaks out of the block early, skipping further actions. If false, the program continues to execute sequentially, updating a memory buffer with the value -1. Figure 5c illustrates this conditional branching with red and green arrows, highlighting the varying execution paths that can lead to timing differences. In the linearized version (Figure 5b), these sensitive-data-dependent branches are replaced by constant-time selectors, ensuring that the execution time remains the same regardless of input values, thereby eliminating timing side channels. Specifically, lines 6 to 9 in Figure 5b show the constant-time selector controlling the memory update. If $sens_data is false, the memory buffer is updated with -1. If true, the original value in memory remains unchanged. Figure 5d shows the revised control flow where the program now consistently follows a single path. This approach ensures that the program's behavior mimics the original logic but without the timing discrepancies that would expose sensitive data through side channels.

In Section 4, We will discuss our detailed methodology for repairing instruction-timing side channels in WebAssembly programs.

## 3 Threat Model

Our threat model focuses on instruction-timing side-channel vulnerabilities in WebAssembly caused by branches that depend on sensitive data, such as secret passwords or cryptographic keys. Other types of side-channel attacks, including cache leaks, Spectre attacks, power side-channel leaks, and other microarchitectural vulnerabilities, are beyond the scope of this work. Additionally, we limit our scope to leaks originating within WebAssembly modules and assume no side-channel vulnerabilities exist in the host environment. The attacker is presumed capable of executing the WebAssembly program repeatedly to gather timing data using various timing strategies in different granularities – ranging from JavaScript APIs like `performance.now()` to CPU cycle-accurate timers such as `rdtscp` – to infer sensitive information, though they cannot directly access the program's variables or memory.

## 4 Design of WASCR

Figure 1 illustrates WASCR's architecture. It takes a WebAssembly module in WebAssembly Text Format (WAT), along with user-annotated sensitive data and a list of function names, as inputs. WASCR consists of two main components: *Leakage Detection* and *Rule-based Code Transformation*. First, it uses WABT [57], an open-source WebAssembly binary toolkit, to parse the WAT module and construct a Program Dependency Graph (PDG). Through a sound static taint analysis, WASCR traces data and control dependencies to identify blocks with conditional branches that depend on sensitive data, which may cause instruction-timing side-channel leaks. In this context, "block" refers to WebAssembly structures such as `block`, `if`, and `loop` instructions containing conditional branches. Next, WASCR repairs these vulnerable branches by applying rule-based code transformations, employing constant-time selectors on the Abstract Syntax Tree (AST) of the WebAssembly module to eliminate timing discrepancies. The modified AST is subsequently converted back to WAT format, completing the repair process.

### 4.1 Leakage Detection

WASCR detects instruction-timing side-channel vulnerabilities by identifying conditional branches dependent on sensitive data and their associated blocks. This begins by constructing a PDG for the WebAssembly module, followed by taint analysis on the graph.
**Program Dependency Graph.** The PDG for a given WebAssembly module consists of four key components: Abstract Syntax Tree (AST), Control Flow Graph (CFG), Control Dependency Graph (CDG), and Data Dependency Graph (DDG). By leveraging control and data dependencies, we propagate sensitive annotations across the PDG, tainting all nodes that handle sensitive data. This process identifies sensitive branches and blocks that can be exploited as side channels, indicating where repairs are needed.
**Sound Static Analysis.** Once PDG is built and sensitive data is annotated, WASCR propagates these sensitive annotations through the data and control dependency edges to identify all sensitive branches and their corresponding blocks needing repair.

```
1  (global $__stack_pointer (mut i32) (i32.const 65536))
2  (func $mem_example (param $key i32)
3      (local $addr1 i32) (local $addr2 i32)
4      local.get $addr1
5      local.get $key
6      i32.store offset=0;; store key to memory
7      ...
8      local.get $addr2
9      i32.load offset=0;; load data, probably get the key
10 )
```

**Figure 6: Challenge in handling memory access operations**

**Algorithm 1** Sound Static Taint Analysis

1: **function** DETECTSENSITIVEBLOCK($S$, $op$)
2:     let $stack = S$
3:     **while** $stack$ is not empty **do**
4:         let $s$ = stack.top()
5:         stack.pop()
6:         **if** $op$ is true and $s$ represents a newly traced function $F$ **then**
7:             push all memory load instructions in $F$ to stack
8:         **end if**
9:         **for all** $s'$ is data/control dependent on $s$ **do**
10:            **if** $s'$ was pushed into $stack$ **then**
11:                continue
12:            **end if**
13:            stack.push($s'$)
14:            **if** $s'$ is a conditional branch instruction **then**
15:                mark the corresponding block as requiring program repair
16:            **end if**
17:        **end for**
18:    **end while**
19: **end function**

Algorithm 1 outlines the taint analysis process, which identifies all sensitive branches and blocks that require repair. For a given WebAssembly module, we begin by initializing a stack with a list of sensitive variables. This stack is maintained to manage nodes during propagation. We traverse the neighboring nodes of the top node on the stack, following data and control dependency edges. If a node has not been visited, it is pushed onto the stack for further propagation. When encountering conditional branch instructions, such as `if` and `br_if`, we mark their associated blocks for repair.

A key challenge arises during propagation when handling memory access operations (e.g., `i32.store` and `i32.load`), which can create implicit data dependencies through WebAssembly's linear memory. For example, in Figure 6, the sensitive key stored in linear memory via `i32.store` (line 6) can later be retrieved by `i32.load` (line 9) if the memory address variables ($addr1 and $addr2) resolve to the same value at runtime, even though there are no direct data or control dependencies between these instructions.

Despite prior efforts on points-to analysis in other languages [7, 9, 39, 49], applying these techniques to WebAssembly is challenging due to its unique memory access patterns. Instead of traditional pointers, WebAssembly uses an offset-based memory access system, where memory operations can access any valid address within linear memory at runtime. In some of our dataset samples, memory addresses are calculated using offsets added to the global variable $__stack_pointer, which often leads to overestimation. These factors make points-to analysis more complex in WebAssembly.

To address this, WASCR adopts a conservative approach. By default, implicit data dependencies between memory store and load instructions are matched based on their address variables and offsets. This approach serves as the baseline configuration in our experiments. Additionally, we offer an option for users to treat all memory access operations within traced sensitive functions as sensitive. When taint propagation enters a new function, WASCR

```
1  local.get $cond1
2  if $I0
3      ...
4      block $B0
5          local.get $cond2
6          br_if $B0
7          local.get $update
8          local.set $val
9      end
10 end
```

(a) Original code

```
1  local.get $update
2  local.get $val
3  local.get $cond1
4  local.get $cond2
5  i32.eqz
6  i32.and
7  select
8  local.set $val
```

(b) Transformed code

**Figure 7: Example of nested branches using condition list**

automatically adds all memory load instructions within that function to the tracing stack.

## 4.2 Rule-based Code Transformation

After identifying sensitive nodes and their associated blocks, WASCR implements a rule-based code transformation to mitigate instruction-timing side channels. This transformation has two main goals: (1) to eliminate all conditional branches within marked sensitive blocks, and (2) to ensure that program semantics remain correct after linearization.

The removal of conditional branches is straightforward, which involves eliminating all branch instructions and any related sensitive block structures. However, as this step alone would break the original program's semantics, additional code transformations are required to ensure the correctness of execution.

**Table 1: Basic store & set transformation rules**

| Operations | Original Instructions | Repaired Version |
|---|---|---|
| Variable set | local.get $cond
if
    ...
    local.set $p
end | local.get $temp
local.get $p
local.get $temp
local.get $cond
select
local.set $p |
| Memory store | local.get $cond
if
    ...
    local.get $addr
    local.get $val
    i32.store
end | local.get $addr
i32.load
local.set $prev_val
local.get $addr
local.get $prev_val
local.get $val
select
i32.store |

**Basic Transformation Rules.** Most WebAssembly instructions do not require modification when conditional branches are removed, as they only affect the program implicitly through the WebAssembly virtual stack machine (e.g., `i32.add`). Our main focus is ensuring memory and variable states are maintained to preserve program correctness. Thus, our transformations specifically target memory store and variable set operations. Table 1 outlines the basic transformation rules (excluding instructions with similar logic, such as `local.tee` and `i64.store`). For each `store` and `set` instruction within a conditional branch, the execution is governed by the associated condition. To maintain correct semantics, we use the WebAssembly constant-time `select` instruction to choose between the updated and original values based on the condition.

**Nested Branches.** In nested conditional branches, instructions can be influenced by multiple conditions. To address this complexity, we generate a condition list for each instruction. This list acts as a synthetic condition that provides a generalized representation of the execution context. As shown in Figure 7a, all instructions

```
1  block $B1
2    loop $L2
3      local.get $loop_cond
4      br_if $B1
5      local.get $break_cond
6      br_if $B1
7      ...
8    end
9  end
```

(a) Loop in WebAssembly

```
1  for(int i=0; i<L; i++){
2      if (break_cond) {
3          break;
4      }
5  }
```

(b) Loop in C

**Figure 8: Loop structure in WebAssembly v.s. C**

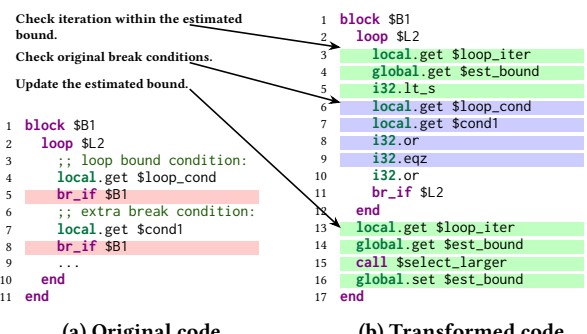

(a) Original code    (b) Transformed code

**Figure 9: Example of loop transformation**

within the inner block $B0 are governed by both the `br_if` and outer `if` conditions. For `if` blocks, we store their condition values and add them to the condition list for all enclosed instructions. For `br_if`, we append the inverse of the branch condition to the condition list for all subsequent instructions in the breaking block, as those instructions will be skipped if the condition evaluates to true. For example, in Figure 7a, the `local.set $val` instruction is controlled by both the `if` condition ($cond1) and `br_if` condition ($cond2). Consequently, in the transformed program ( Figure 7b), `local.set $val` will execute only if $cond1 is true and $cond2 is false, based on the condition list derived from these two conditions. A special case is the `br` instruction. We convert all `br` instructions to `br_if` instructions with true conditions for consistency.

**Loop Iterations.** A key challenge in constant-time program repair is managing loop iterations when the loop upper bound is deemed sensitive. To ensure execution time remains independent of sensitive data, it is crucial to establish a fixed loop upper bound. Existing approaches either statistically estimate upper bounds [48, 59] or dynamically manage iteration numbers in Just-in-Time environments [52], both with limitations: the former lacks flexibility, while the latter is unavailable in our environment. Given these constraints, we propose using Large Language Models (LLMs) to analyze the loop and estimate a preliminary upper bound, followed by static code transformation to adaptively adjust this upper bound during loop execution.

While WebAssembly loops share similarities with those in other languages, they exhibit unique behaviors. As shown in Figure 8, a single `break` statement in C may translate to multiple `br_if` instructions in WebAssembly. These `br_if` instructions collectively serve to exit the same loop, which complicates static analyses to identify the loop's iteration bound exit condition. To address this issue, our

```
1  (func $Callee (param $p i32)
2    global.get $global_cond
3    ...
4  )
5  (func $Caller (param
       ↪ $key_cond i32)
6    (local $l i32)
7    global.get $global_cond
8    local.get $key_cond
9    i32.and
10   global.set $global_cond
11   local.get $l
12   call $Callee
13 )
```

```
1  (func $Callee (param $p i32)
2    ...
3  )
4  (func $Caller (param
       ↪ $key_cond i32)
5    (local $l i32)
6    local.get $key_cond
7    if $I0
8      local.get $l
9      call $Callee
10   end
11 )
```

(a) Function condition - Before    (b) Function condition - After

**Figure 10: Example of function calls in sensitive branches**

approach replaces all breaking points and enforces adherence to our custom upper bound for the WebAssembly loop.

To determine loop upper bound and enhance the soundness of the static analysis for loop handling, we utilize LLMs. Specifically, we input the loop body into ChatGPT [38], prompting it to identify loop iteration conditions and estimate an upper bound. We then remove all `br` and `br_if` instructions from the loop block, incorporating them into the condition lists of enclosed instructions within the loop. Finally, we append the iteration process to the end of the loop structure, adhering to the estimated upper bound.

However, this initial estimate may be overly conservative, allowing the loop to exceed the estimated maximum. To address this, we implement an adaptive bound update mechanism. We store the estimated upper bound in a global variable, which is dynamically updated if the actual number of loop iterations exceeds the current estimate. This global variable persists across multiple executions of the same WebAssembly module, enabling incremental refinement of the upper bound with each run. This iterative process facilitates convergence toward a more accurate upper bound.

Figure 9 illustrates the transformation process. Initially, the `br_if` instructions (Figure 9a) are removed, with their conditions stored in local variables. During each iteration, the value of `$loop_iter` is incremented and compared against the estimated upper bound. The loop terminates only if either `$cond1` or `$loop_cond` is satisfied, and the iteration count exceeds the estimated upper bound (lines 3 to 11, Figure 9b). To ensure correct exit conditions, these conditions are evaluated as true only upon the first execution where they are met. Subsequent executions will not alter their value once the exit decision is made. Finally, the estimated upper bound is updated by comparing it to the actual number of loop iterations, using the larger value (lines 13 to 16, Figure 9b).

**Function Calls in Sensitive Branches.** In cases where function calls occur within sensitive branches, we must ensure these functions are properly linearized. For each function, we use a global variable to manage its execution status. Figure 10 illustrates this approach. Since WebAssembly operates in a single-threaded, sequential environment, the caller can set this global variable to either true or false before invoking a function, indicating whether the function would be called in the original branch. Next, all memory store instructions within the callee function are governed by this condition, in addition to any original constraints. Local variable set instructions within the callee are not considered, as they only affect the callee function and do not impact the outer program.

**Function/Block Return Values.** In WebAssembly, block instructions (such as `if` blocks) can terminate with return values, similar to functions. To ensure consistent timing behavior, we enforce a single exit point per block or function by eliminating all alternative exit paths. To preserve return value semantics during transformation, we temporarily store the return value and use the `select` instruction at the end of each block or function to determine and return the correct value, preserving the original behavior.

**Indirect Function Calls and Break Tables.** WebAssembly's indirect function calls are based on runtime indexes, allowing the selection of the appropriate function at runtime. To ensure accurate static analysis and facilitate code transformation, all indirect function calls are converted into direct calls accompanied by multiple conditional branches, each corresponding to a potential function that matches the signature. Similarly, break tables are transformed into multiple `br_if` instructions.

## 4.3 Correctness Analysis

We claim that our methodology ensures the correctness of repairing instruction-timing side channels in terms of both *leakage detection* and *code transformation*.

**Soundness of Leakage Detection.** Our side-channel detection employs a sound taint analysis approach. We ensure that taint propagation comprehensively covers all possible nodes that are either control- or data-dependent on sensitive information. Although this process may over-approximate, it guarantees that no branches dependent on sensitive data are overlooked. Therefore, this thorough coverage effectively ensures no instruction-timing side channels remain after the subsequent code transformation phase.

**Correctness of Code Transformation.** Our code transformation preserves the semantic equivalence of the original program through a meticulously designed linearization process. This process ensures that variable set and memory store instructions in the repaired programs take effect only when all associated conditions are satisfied, thereby guaranteeing that the memory and variable states remain consistent with those of the original programs.

## 5 Evaluation

In this section, we evaluate WᴀSCR's ability to repair instruction-timing side-channel leaks in WebAssembly modules. Our evaluation is guided by three key research questions that focus on effectiveness, efficiency, and quality of the repaired programs:

- **RQ1 (Effectiveness of WᴀSCR)**: How effectively does WᴀSCR repair instruction-timing side channels in WebAssembly modules?
- **RQ2 (Efficiency of the Repair Process)**: How quickly does WᴀSCR complete the repair process?
- **RQ3 (Quality of the Repaired Programs)**: What is the quality of the repaired programs in terms of execution time and code size?

## 5.1 Experiment Setup

We conduct our experiments on the GEM5 simulator, which allows us to customize CPU components to isolate cache miss interference, ensuring evaluation accuracy and fairness. To execute the

WebAssembly modules, we use WasmEdge [2] as the runtime environment within GEM5, employing the WasmEdge C API to load and interact with WebAssembly modules, while compiling the C host program to x86 binary for GEM5 simulation. This setup enables precise measurement of CPU cycles specifically for the tested WebAssembly functions, yielding deterministic results.

Our evaluation dataset consists of 20 samples in total. We collected 12 samples from previous studies on instruction-timing attacks, including Wu et al. [59], Sorares et al. [47], Disselkoen et al. [20], and Borrello et al. [12], which assess side-channel elimination. These sources, originally written in C/C++, were compiled to WebAssembly using Emscripten [1], the most widely used WebAssembly compiler [44]. Additionally, we selected 8 real-world WebAssembly modules collected by Romano et al. [45] and Hilbig et al. [27]. To adapt the samples for testing with WasmEdge, we made minor modifications to export core functionalities of these WebAssembly modules (e.g., encryption) to the host environment, preserving key functionalities while facilitating evaluation.

**Table 2: GEM5 simulation results of WASCR**

| Program | Before Fixing | | | After Fixing | | |
| --- | --- | --- | --- | --- | --- | --- |
| | Input 1 | Input 2 | \|Delta\| | Input 1 | Input 2 | \|Delta\| |
| des [46] | 12362766 | 14227470 | 1864704 | 20034520 | 20034520 | 0 |
| loki91 [46] | 24024119 | 23540603 | 483516 | 130371440 | 130371440 | 0 |
| 3way [46] | 1684764 | 1718896 | 34132 | 2429415 | 2429415 | 0 |
| twofish [30] | 21018532 | 20944796 | 73736 | 32466778 | 32466778 | 0 |
| tls-rempad-luk13 [19] | 585945 | 95405 | 490540 | 846894 | 846894 | 0 |
| findmax [40] | 2593284 | 1926951 | 666333 | 3910999 | 3910999 | 0 |
| binsearch [40] | 76556 | 86159 | 9603 | 107105 | 107105 | 0 |
| histogram [40] | 5400615 | 5572615 | 172000 | 9679317 | 9679317 | 0 |
| rsort [40] | 2457558 | 3970219 | 1512661 | 23719489 | 23719489 | 0 |
| hash-one [47] | 863457 | 594401 | 269056 | 1672259 | 1672259 | 0 |
| plain-many [47] | 420241 | 43313709 | 42893468 | 286627044 | 286627044 | 0 |
| check_password [20] | 167113 | 81354 | 85759 | 446449 | 446449 | 0 |
| xsalsa20_xor [45] | 13194536 | 121603 | 13072933 | 17642213 | 17642213 | 0 |
| process [45] | 77706 | 78298 | 592 | 187394 | 187394 | 0 |
| thinning_zs [27] | 708776 | 102373 | 606403 | 1726690 | 1726690 | 0 |
| hyphenate [27] | 146883 | 79964 | 66919 | 517213 | 517213 | 0 |
| rotate [27] | 211958 | 280070 | 68112 | 16987167 | 16987167 | 0 |
| sha256_bench [27] | 210639 | 207399 | 3240 | 1036356 | 1036356 | 0 |
| test [27] | 163786 | 160720 | 3066 | 820504 | 820504 | 0 |
| sha1_bench [27] | 238139 | 235292 | 2847 | 2397430 | 2397430 | 0 |

## 5.2 RQ1: Effectiveness of WASCR

To evaluate the effectiveness of WASCR, we manually analyzed the C/C++ source code and created two distinct inputs for each WebAssembly function, designed to trigger different execution paths and resulting in varying execution times. For the real-world WebAssembly samples, randomly chosen inputs were utilized. For each sample, we measured the CPU cycles of the WebAssembly function executed within GEM5, both before and after applying WASCR, using the designed inputs. To isolate the influence of adaptive loop management and runtime environment, we warm up the tested WebAssembly functions in advance with both inputs. This pre-execution phase establishes consistent loop bounds, ensuring reliable CPU cycle measurement.

Table 2 presents the results, showing that without WASCR, the CPU cycles for each sample vary between the two designed inputs, indicating vulnerability to instruction-timing side-channel attacks. In contrast, after applying WASCR, such timing variances are eliminated, thereby mitigating potential leaks from timing attacks.

**RQ1 Takeaway:** WASCR effectively repairs instruction-timing side channels, enhancing WebAssembly security against such leaks.

## 5.3 RQ2: Efficiency of the Repair Process

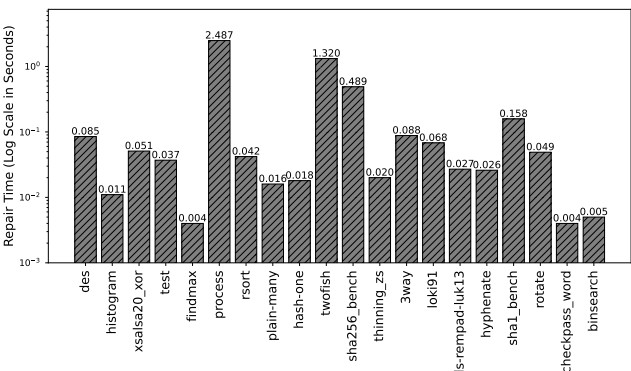

**Figure 11: Program repair time of WASCR**

We measured the time WASCR took to repair each WebAssembly module, encompassing both leakage detection and code transformation phases. The results, averaged over ten repair executions, are presented in Figure 11. Our findings show that WASCR typically completes repairs within a few seconds, demonstrating efficiency for practical use.

**RQ2 Takeaway:** WASCR efficiently repairs the selected samples, completing all tasks within a few seconds, thereby demonstrating its practicality.

## 5.4 RQ3: Quality of the Repaired Programs

WASCR introduces overhead due to code transformation, which linearizes sensitive branches and manages loop bounds. We measured its overhead in terms of runtime performance and code size increase to assess the quality of repaired programs.

**Runtime overhead.** To illustrate the runtime overhead introduced by WASCR, we calculated the ratio of the CPU cycles for each WebAssembly module after applying WASCR to the larger CPU cycle count of the two inputs before fixing, as shown in Table 3. The results indicate that WASCR can introduce runtime overhead through code transformation. We consider the overhead of most samples to be acceptable, as branch linearization requires executing additional paths not present in the original implementations. In contrast, for some real-world WebAssembly modules, we mark all elements as sensitive to reduce manual effort and demonstrate the robustness of our approach without knowing each function's purpose. This forces all branches to be transformed and linearized, resulting in higher overhead. For example, in the rotate sample, we manually confirmed that it performs image rotation. Linearization forces the program to handle all possible sizes and angles, leading to an overhead increase of approximately 60×.

**Code size increase.** We compared the lines of code for each WebAssembly module before and after applying WASCR, calculating the ratio as presented in Table 3. The code size increase generally aligns with the runtime overhead, which we also deem acceptable.

**RQ3 Takeaway:** WₐSCR generally produces repaired programs of good quality on most samples, balancing enhanced protection with moderate performance and code size overhead.

**Table 3: Code size increase and runtime overhead**

| Program | Code Size | | | Runtime Overhead |
|---|---|---|---|---|
| | LoC - Origin | LoC - Repaired | Overhead | |
| des | 1637 | 2510 | 1.41x | 1.53x |
| loki91 | 1710 | 3508 | 5.43x | 2.05x |
| 3way | 2345 | 3395 | 1.41x | 1.45x |
| twofish | 13765 | 22866 | 1.54x | 1.66x |
| tls-rempad-luk13 | 195 | 504 | 1.44x | 2.58x |
| findmax | 205 | 348 | 1.51x | 1.70x |
| binsearch | 74 | 190 | 1.24x | 2.57x |
| histogram | 285 | 485 | 1.79x | 1.70x |
| rsort | 876 | 2320 | 5.97x | 2.64x |
| hash-one | 378 | 733 | 1.94x | 1.94x |
| plain-many | 322 | 1330 | 6.61x | 4.13x |
| check_password | 216 | 661 | 2.67x | 3.06x |
| xsalsa20_xor | 1211 | 2042 | 1.68x | 1.33x |
| process | 9172 | 10905 | 1.18x | 2.39x |
| thinning_zs | 346 | 787 | 2.27x | 2.43x |
| hyphenate | 529 | 997 | 1.88x | 3.52x |
| rotate | 748 | 5354 | 7.13x | 60.6x |
| sha256_bench | 4169 | 12930 | 3.10x | 4.92x |
| test | 1101 | 3978 | 2.67x | 5.01x |
| sha1_bench | 2165 | 11003 | 5.08x | 10.06x |

## 6 Related Works

**Instruction-timing Side-Channel Attacks and Mitigation.** Numerous studies have focused on detecting and mitigating timing side-channel attacks. Geimer et al. [22] provide a comprehensive survey of timing side-channel detectors. Many of these tools employ static verification methods, such as static taint analysis [15, 20, 47, 59], symbolic execution [19, 20], and other static methods [5, 6, 8, 42], to identify potential side channels in programs. Another direction involves dynamic detection using techniques like fuzzing [25, 37], statistical testing [41], etc. These approaches have demonstrated effectiveness in side channel verification.

Linearizing sensitive branches through program transformation is an effective method to eliminate instruction-timing side channels. Wu et al. [59] applied static detection and constant-time selectors to mitigate these leaks in C/C++ programs. As follow-up works, Soares et al. [48] extended this approach with additional bound checks for memory safety, while Borrello et al. [12] utilized an adaptive just-in-time strategy to handle loop iterations, which we follow in our study. Under the Just-in-Time (JIT) environment, Cleemput et al. [52] proposed a dynamic approach to repair the leaking code.

While these approaches effectively eliminate timing side channels in their respective contexts, they do not target WebAssembly-specific side channels, which is the focus of our research.

**Side-Channel Attacks in WebAssembly.** Despite incorporating several security mechanisms, WebAssembly remains susceptible to side-channel attacks [3, 28, 55, 56], particular instruction-timing attacks, which continue to be a significant concern within the WebAssembly community. Current research mainly focuses on statically verifying the constant-time property of WebAssembly modules. For instance, Watt et al. [54] proposed CT-Wasm, an extension

of WebAssembly's types and semantics to verify the constant-time property concerning sensitive data. Tsoupidi et al. [51] employed a Relational Symbolic Execution (RelSE) based approach to discover constant-time violations in WebAssembly modules. However, these works require modifications to WebAssembly runtimes or interpreters and do not provide automatic fixing for addressing potential side channels. While other studies have explored protections against microarchitecture side channels, such as cache attacks [14] and speculative execution leaks [36, 53], they do not provide a safeguard regarding instruction-timing side channels.

In contrast, our approach provides a platform-independent approach for automatic detection and repair of instruction-timing side channels, effectively filling this gap in existing research.

## 7 Discussion

In this section, we discuss our current limitations and future directions to improve our work.

**Imported Function Calls.** WebAssembly modules can import and execute external functions (e.g., JavaScript APIs), which may be invoked within sensitive code branches. One potential solution is to expand the trace path into these imported functions and introduce an additional parameter to indicate whether the functions should execute, allowing for behavior adjustments. However, implementing this approach would require modifications to the runtime environment, which is beyond the scope of our current work. We have identified this as an area for future improvement.

**Multidimensional Side Channels.** Our work exclusively focuses on WebAssembly instruction-timing side channels, while other types of side-channel attacks, such as microarchitecture attacks, are beyond its scope. Additionally, the current WebAssembly specification does not mandate the `select` instruction to be translated into constant-time machine code (e.g., `CMOV` on x86 or ARM), thus WebAssembly runtimes could implement this translation differently. Although our code inspections and GEM5 simulations confirm that today's WebAssembly runtimes typically provide this guarantee, ensuring this property consistently would require collaboration with the WebAssembly standardization committee to incorporate a constant-time requirement into the official specification. We consider this a potential work for future research.

## 8 Conclusion

In this paper, we introduce WₐSCR, an automated WebAssembly instruction-timing side channel repairer. WₐSCR employs static code transformation to protect WebAssembly from timing side-channel vulnerabilities. Through carefully designed leakage detection and transformation rules, we achieve full linearization of control flows, providing a robust and compatible approach that supports various WebAssembly runtimes. We demonstrate that WₐSCR effectively mitigates instruction-timing side channels, achieving moderate repair time and high repair quality.

## 9 Data Availability

Our code is available at https://anonymous.4open.science/r/Depgraph-wasm-29E0/.

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

Received 20 February 2007; revised 12 March 2009; accepted 5 June 2009

