# OpenReview forum: "WaSCR: A WebAssembly Instruction-Timing Side Channel Repairer"
_ACM.org/TheWebConf/2025/Conference — WWW 2025 Poster_

### Official Review · Reviewer_Rjyd · 2024-11-23

**Novelty:** 1
**Technical Quality:** 1

**Review:**

The authors propose a tool named WaSCR designed to detect and repair instruction-timing side channel vulnerabilities in WebAssembly programs. Utilizing static analysis and rule-based code transformations, WaSCR is able to effectively identify and remediate parts of the code that may lead to information leakage. While this work demonstrates innovation in technical implementation and theoretical analysis, it exhibits some deficiencies in the depth of innovation, breadth of experiments, and comparison with existing technologies.

**Questions:**

1.	The author contends that existing methods such as those referenced in [51, 54] necessitate platform-specific extensions and involve substantial manual effort to amend violations. This assertion that current solutions demand extensive manual labor may be exaggerated or might require a more nuanced interpretation. From my perspective, CT-Wasm effectively categorizes data into public and secret types, leveraging its type system to automatically enforce security policies. This architectural approach inherently reduces the likelihood of timing attacks, thereby diminishing the necessity for manual intervention during the coding phase. The motivation behind the proposed solution in this paper is to address the drawbacks of strong platform dependency, poor scalability, and high manual effort required by existing approaches, claiming that the proposed solution can effectively overcome these deficiencies. However, the author's description of these aspects in the manuscript is notably brief and vague, complicating the assessment of the paper's value.
The author must provide a detailed account of the limitations of schemes like [51, 54] in terms of platform extension and automation, with specific examples to substantiate their claims.
2. Regarding the design of WaSCR, I do not consider the rule-based code transformation module to be particularly advanced. Fixed rules may fail to address emerging or unforeseen side-channel patterns, necessitating manual updates to the transformation rules. This rigidity could potentially limit WaSCR's applicability in more dynamic or evolving environments. Moreover, I believe that WaSCR’s reliance on predefined rules restricts its ability to handle edge cases or complex code constructs. For instance:
1）Function calls within sensitive branches necessitate the use of external global variables for tracking, which complicates transformations.
2）Indirect function calls are converted to multiple br_if conditions, which not only increases the code size but also introduces potential maintenance challenges.
Adding to the point on maintenance, rule-based systems inherently demand frequent updates to remain effective as new side-channel vulnerabilities or WebAssembly features emerge. This increases the maintenance overhead and makes the system less future-proof compared to adaptive methods, such as those employing machine learning-based approaches. This aspect should be carefully considered when evaluating the long-term viability and scalability of WaSCR.
3. The paper mentions experiments conducted on 20 WebAssembly modules, demonstrating WaSCR's capability in repairing instruction-timing side channels. However, these experiments lack direct comparisons with other existing technologies. Without such comparisons, it is challenging to assess the advantages and practical application value of WaSCR. Additionally, the discussion of experimental results primarily focuses on the performance overhead post-repair, with insufficient evaluation of security enhancement and actual protective effects.
4. The paper may not have comprehensively evaluated the security of programs post-repair. It remains uncertain whether all types of instruction-timing side channels have been effectively identified and remedied.

**Reviewer Confidence:**

4: The reviewer is certain that the evaluation is correct and very familiar with the relevant literature

**Scope:**

3: The work is somewhat relevant to the Web and to the track, and is of narrow interest to a sub-community

---

### Official Review · Reviewer_KJZG · 2024-11-29

**Novelty:** 5
**Technical Quality:** 5

**Review:**

Dear authors, thanks for submitting your work to The Web Conf 2025.

This paper discusses instruction timing side channel attack that might exist in Wasm applications, and proposes techniques to detect and repair related vulnerabilities. Overall, the paper is well written. I do appreciate the Background and Motivation section, which gives very thorough explanation on the domain specific features of WebAssembly, the definition of instruction-timing side channels, and how to repair such side channels with constant time selectors. That being said, I do have a few suggestions listed as below:

1. Motivation. What are some concrete evidences that Wasm might be attacked by instruction-timing side channels? It is unclear whether creating such an attack is practical or not, given the randomness caused by all types of interferences (e.g. non-Wasm components, cache misses). The motivation would be greatly enhanced with a real world example showing e.g. exposing some password using instruction-timing side channels.

2. Technical solution. The paper builds on the assumption that if the instruction timing delta among all possible inputs is zero, then there is no way for attackers to build a side channel. While this is correct, I'm not sure whether this is necessary. A strawman solution, for example, would be to slightly randomize/obscure the execution time, so that attackers cannot retrieve useful information based on instruction timing. This strawman solution may also present smaller LoC/time overhead, thus more practical in real world settings. Of course, this may not be the best strawman solution out there, but the point is, why do the authors believe constant-time is the best way to go in this context?

3. Evaluation. My concern on evaluation is largely similar to my questions on solution. What are some other baselines other than constant-time code transformation? Is there a more fine-grained tradeoff between overhead and level of protection?

4. Presentation. The paper rely heavily on binary code snippets, many of which I find hard to understand. Figure 5 (c) and (d) instead visualized the control flow, which is much easier to understand. Perhaps also consider replacing other code snippets with better visualization?

Please address/clarify these questions during the next few iterations. I'm looking forward to response from the authors.

**Questions:**

Most of my questions are covered in the review, e.g.
1. What are some concrete evidences that Wasm might be attacked by instruction-timing side channels?
2. Why do the authors believe constant-time is the best way to fix instruction-timing side channels in Wasm applications?
3. What are some other baselines other than constant-time code transformation? Is there a more fine-grained tradeoff between overhead and level of protection?
4. What are some better ways to present technical details in the solution section?

**Reviewer Confidence:**

2: The reviewer is willing to defend the evaluation, but it is likely that the reviewer did not understand parts of the paper

**Scope:**

3: The work is somewhat relevant to the Web and to the track, and is of narrow interest to a sub-community

---

### Official Review · Reviewer_qLg7 · 2024-12-02

**Novelty:** 5
**Technical Quality:** 6

**Review:**

## Summary
Existing approaches for mitigating instruction timing side-channel vulnerabilities are platform-dependent and require significant manual effort from developers. Furthermore, these methods are not portable to WebAssembly (Wasm) due to its unique system design. To address this, WaSCR introduces an automated WebAssembly instruction timing side-channel repair mechanism.

WaSCR’s primary concept is to linearize conditional blocks in Wasm and enforce fixed loop iterations. The assumption is that sensitive information (e.g., passwords) may be matched during loop iterations with conditional blocks, and attackers could exploit timing side channels to infer this information. WaSCR detects vulnerabilities by analyzing conditional branches that depend on sensitive data (Leakage Detection). To mitigate these vulnerabilities, WaSCR transforms code using rules that linearize conditional blocks and enforce fixed loop iterations regardless of input differences (Rule-based Code Transformation).

WaSCR claims that its approach effectively detects all branches dependent on sensitive data and ensures that its transformations preserve the semantics of the original program. The paper demonstrates that WaSCR is effective, efficient, and produces high-quality results.

## Strengths
- The paper is well-written and easy to understand.
- Experiments show that the proposed technique effectively mitigates timing side-channel attacks on Wasm code.

## Weaknesses
- Concerns regarding compatibility with various Wasm JIT compilers.
- Lack of detailed explanation of the design.
- Insufficient detailed analysis of experimental results.

## Comments
As Wasm is a type of bytecode format, distributed Wasm code can be processed in various ways by Wasm runtimes, such as through interpreter, JIT, or AOT modes. However, the proposed technique assumes that Wasm code is JIT-compiled into machine code. It also presumes that the given JIT compiler translates Wasm select instructions into constant-time machine code (e.g., CMOV). While the authors acknowledge these limitations in section 7, such assumptions undermine the claim that WaSCR is platform-independent.

While the introduction briefly highlights the challenges of repairing instruction-timing side channels due to Wasm’s unique characteristics, the Design section lacks detailed explanations of these challenges and how they were addressed. For instance, handling memory access operations is one such area.

Additionally, some experimental results would benefit from more in-depth analysis (refer to Questions for details).

**Questions:**

- A more detailed explanation is needed regarding the statement: 'To address this, WaSCR adopts a conservative approach. By default, implicit data dependencies between memory store and load instructions are matched based on their address variables and offsets.'

- Table 3 shows that rotate exhibits a significant performance overhead of 60.6x. What are the underlying reasons for this overhead?

- Does WaSCR also support cases where sensitive data is stored in global variables rather than in linear memory?

**Reviewer Confidence:**

3: The reviewer is confident but not certain that the evaluation is correct

**Scope:**

3: The work is somewhat relevant to the Web and to the track, and is of narrow interest to a sub-community

---

### Official Review · Reviewer_ABnH · 2024-12-03

**Novelty:** 5
**Technical Quality:** 6

**Review:**

Summary:
The paper introduces WaSCR, a tool for automatically detecting and repairing instruction-timing side-channel vulnerabilities in WebAssembly programs. By leveraging a static analysis approach, WaSCR uses program dependency graphs and taint analysis to identify timing leaks and applies rule-based transformations to ensure execution independence from sensitive data. The authors present a clear problem definition, summarize existing approaches, and move from a high-level overview to implementation details of the tool. The evaluation demonstrates WaSCR effectiveness in eliminating timing side channels while maintaining program correctness with acceptable performance overheads.

Strengths:
- The introduction provides a clear definition of the problem, existing challenges, and motivation.
- The paper is well-written, with a logical flow from problem definition to solution and evaluation.
- Results from diverse benchmarks demonstrate the tool's applicability and efficiency.

Weaknesses:
- The focus on instruction-timing side channels leaves other WebAssembly vulnerabilities unexplored.
- The tool requires annotated sensitive data, potentially limiting usability in fully automated workflows.
- The paper has limited discussion of alternative approaches, such as runtime-based mitigation techniques, and lacks a detailed comparison with such tools.

**Questions:**

- Could you explain why WaSCR is unable to support other types of side-channel attacks, such as cache leaks, Spectre attacks, and other microarchitectural vulnerabilities, to better understand the limitations of the approach and possibilities for tool extensions?
- Section 7 mentions the limitation of executing external functions within sensitive code branches. What is the prevalence of such functions in the considered code samples? Does WaSCR simply ignore such branches, thereby leaving the code vulnerable?
- The code transformations in Figure 5(b) introduce `$stack_ptr`. Is this a typo? If not, could you explain this transformation in more detail?

**Reviewer Confidence:**

3: The reviewer is confident but not certain that the evaluation is correct

**Scope:**

3: The work is somewhat relevant to the Web and to the track, and is of narrow interest to a sub-community